# Hierarchical Reinforcement Learning Framework in Geographic Coordination for Air Combat Tactical Pursuit

**DOI:** 10.3390/e25101409

**Published:** 2023-10-01

**Authors:** Ruihai Chen, Hao Li, Guanwei Yan, Haojie Peng, Qian Zhang

**Affiliations:** 1School of Aeronautics, Northwestern Polytechnical University, Xi’an 710072, China; crh@mail.nwpu.edu.cn (R.C.); sc4979@163.com (H.P.); 2Chengdu Aircraft Design and Research Institute, Chengdu 610041, China; 3School of Aerospace, Northwestern Polytechnical University, Xi’an 710072, China

**Keywords:** hierarchical reinforcement learning, meta-learning, reward design, decision

## Abstract

This paper proposes an air combat training framework based on hierarchical reinforcement learning to address the problem of non-convergence in training due to the curse of dimensionality caused by the large state space during air combat tactical pursuit. Using hierarchical reinforcement learning, three-dimensional problems can be transformed into two-dimensional problems, improving training performance compared to other baselines. To further improve the overall learning performance, a meta-learning-based algorithm is established, and the corresponding reward function is designed to further improve the performance of the agent in the air combat tactical chase scenario. The results show that the proposed framework can achieve better performance than the baseline approach.

## 1. Introduction

The application of reinforcement learning (RL) [1,2] in aerial combat has attracted a lot of attention in recent years, and RL has been used to simulate the behavior of pilots and aircraft and to optimize aerial combat strategies [3,4].

Challenges related to these simulations include establishing the interaction model between pilots and aircraft [5,6]; simulating the behavior of pilots maneuvering the aircraft and its impact [7]; introducing enemy aircraft and weapons; simulating the behavior of the enemy aircraft and its impact [8]; and the simulation of multi-aircraft cooperative combat behavior [9,10]. Of these, confrontation behavior in air combat is complex and variable, with various modes [11], and it is difficult for traditional methods such as state machines and differential games to completely characterize the real-time decision-making state of pilots and devise further optimization according to different situations [12,13]. However, by modeling the air combat process as a Markov process [14], reinforcement learning methods can achieve continuous optimization of decision-making algorithms [15,16].

The first application of RL in aerial combat was proposed by Kaelbling et al. [17]. They proposed a model-based RL approach for controlling an unmanned aerial vehicle (UAV) in a simulated air-to-air combat environment. The UAV was equipped with a simulated radar and missile system, and the RL agent was trained to select the optimal action for the UAV to maximize its chances of survival. The results showed that the RL agent was able to outperform the baseline agent in terms of survival rate. More recently, Hu et al. [18] trained long and short-term memory (LSTM) in a deep Q-network (DQN) framework for air combat maneuvering decisions, and this was more forward-looking and efficient in its decision-making than fully connected neural-network- and statistical-principle-based algorithms [19]. In addition, Li proposed a deep reinforcement learning method based on proximal policy optimization (PPO) to learn combat strategies from observation in an end-to-end manner [20,21], and the adversarial results showed that his PPO agent can beat the adversary with a win rate of approximately 97%. Based on the deep deterministic policy gradient algorithm framework, Lu designed and implemented an air warfare decision policy and improved the efficiency of the training process via a preferred experience playback strategy [22]. This method was able to achieve fast convergence while saving training costs.

Because of the sparse nature of the air combat environment, the shaping of the reward function has been a key challenge in the application of reinforcement learning to air combat [23,24]. Piao constructed a high-fidelity air combat simulation environment and proposed a critical air combat event reward-shaping mechanism to reduce episodic win–lose signals [25,26], enabling fast convergence of the training process. The implementation results showed that reinforcement learning can generate a variety of valuable air combat tactical behaviors under beyond-visual-range conditions. Hu et al. [27] designed a reward function based on the original deep reinforcement learning method, and the design dimension of the reward included the real-time gain due to the maneuver as well as the final result gain. For the air combat maneuver decision problem with sparse rewards, Zhan et al. [28,29,30] applied a course-based learning approach to design a decision course of angle, distance, and mixture which improved the speed and stability of training compared to the original method without any course and was able to handle targets from different directions.

In the air combat decision-making process, the combination of various independent states forms a very large situation space which leads to an explosion of state dimensions [31]. Current research focuses on the rationality of the decision logic after the introduction of reinforcement learning training in a specific scenario [32,33], whereas this paper focuses on making the existing decision algorithm rapidly scalable as more and more realistic situations are introduced to quickly adapt to a more realistic air combat countermeasure environment [34,35]. The state space curse of dimensionality problem often leads to insensitivity in the model tracking which eventually fails to converge to a better stable tracking state. Therefore, in this paper, a hierarchical reinforcement learning (HRL)-based air warfare framework is proposed [36], which uses a hierarchical reinforcement learning structure to implement three-dimensional air warfare. Experimental results show that the proposed framework can achieve better performance than existing methods. The main innovations of this study are as follows:We propose a hierarchical reinforcement learning framework in geographic coordination for the training and use of senior and basic policies to solve the MDP in air combat chase scenarios.We propose a meta-learning algorithm applied to the framework proposed in this paper for the complex sub-state and action space learning problem of air warfare. The reward decomposition method proposed in this paper also alleviates the problem of reward sparsity in the training process to some extent.We independently built a three-degrees-of-freedom air combat countermeasure environment and modeled the task as a Markov process problem. Specifically, we defined the key elements of the Markov process, such as state, behavior, and reward functions for this task.We established a quantitative system to evaluate the effectiveness of reinforcement learning methods for training in 3D air combat.

In Section 2, we describe the application of reinforcement learning algorithms to the established air combat environment. In Section 3, we present the algorithm framework, reward function design ideas, algorithm training, and usage process. In Section 4, we establish a standard evaluation method and compare multiple SOTA models. In Section 5, we discuss the experimental results and in Section 6, we summarize the whole paper.

## 2. Reinforcement Learning for Air Combat

This paper sets out a design for a hierarchical RL algorithm capable of learning effective decision strategies in air combat countermeasure scenarios through interaction with a simulated environment. The core of the algorithm is the use of Markov decision processes (MDPs) to model the decision process of combat aircraft in the presence of uncertainty and dynamic adversaries [37,38]. In this context, the design of MDPs requires careful consideration of factors such as state space representation, action selection, and reward function design. In addition, the construction of realistic and challenging combat environments is critical to evaluate the performance of the HRL algorithms constructed in this paper [39,40].

### 2.1. Markov Decision Process

Figure 1 describes the feedback loop; each of the subscripts *t* and *t* + 1 representing a time step refers to a different state: the state at moment t and the state at moment *t* + 1. Unlike other forms of learning, such as supervised and unsupervised learning, reinforcement learning can only be thought of as a series of sequential state–action pairs [41].

Intelligence in reinforcement learning requires information from the current state $s_{t+1}$ and also from the previous state $s_{t}$ to make the best decision that maximizes its payoff [42]. A state signal is said to have Markovianity if it has the information necessary to define the entire history of past states.

Markov decision processes (MDPs) represent decision-makers who periodically observe systems with Markovianity and make sequential decisions [42,43]. They are the framework used for most problems in reinforcement learning. For each state s and action a, the probability that the next state $s^{\prime }$ may occur is
(1)$$P_{ss^{\prime }}^{a}=Pr\left\{s_{t+1}=s^{\prime }|s_{t}=s,a_{t}=a\right\}$$
where P denotes the transfer probability, meaning the possible change of air combat situation when a certain behavior a is executed in a specific state s. In this paper, the value of P is fixed, and the expectation value of the next reward value R can be determined as
(2)$$V^{\pi }(s)=\sum_{a}\;\pi (s,a)\sum_{s^{\prime }}\;P_{ss^{\prime }}^{a}\left[R_{ss^{\prime }}^{a}+\gamma V^{\pi }\left(s^{\prime }\right)\right],\forall s\in S,\forall a\in A$$

Intelligence tries to maximize its payoff over time, and one way to achieve this is to optimize its strategy. A strategy π is optimal when it produces better or equal returns than any other strategy, and π specifies the probability distribution of executing a certain decision action in a given air combat situation. The equation for state values states that at any state, strategy *π* is better than π’ if $V^{\pi }(s)$ ≥ $V^{\pi ’}$, ∀s∈S. The state value function and the state action value function can be optimized according to the following two equations:(3)$$V*(s)=\underset{\pi }{max}\;V^{\pi }(s),s\in S$$
(4)$$Q*(s,a)=\underset{\pi }{max}\;Q^{\pi }(s,a),\forall s\in S,\forall a\in A$$

The above two equations can calculate the optimal state value V*(s) and the optimal action value Q*(s,a) when using the strategy *π*. The Bellman optimal equation for V*(s) can be used to calculate the value of states when the reward function $R_{ss^{\prime }}^{a}$ and the transfer probability $P_{ss^{\prime }}^{a}$ are known without reference to the strategy; similarly, the Bellman optimal equation constructed with the state action value function can be used as follows:(5)$$\begin{array}{c} V^{*}(s)=\underset{a}{max}\;\sum_{s^{\prime }}\;P_{ss^{\prime }}^{a}\left[R_{ss^{\prime }}^{a}+W*\left(s^{\prime }\right)\right]\\ Q^{*}(s,a)=\sum_{s^{\prime }}\;P_{ss^{\prime }}^{a}\left[R_{ss^{\prime }}^{a}+\gamma \underset{a^{\prime }}{max}\;Q^{*}\left(s^{\prime },a^{\prime }\right)\right] \end{array}$$

The above two equations can calculate the optimal state value V*(s) and the optimal action value Q*(s,a) when using the strategy *π*. Additionally, in the case of no reference strategy, when the reward function $R_{ss^{\prime }}^{a}$ and the transfer probability $P_{ss^{\prime }}^{a}$ are known, the Bellman optimal equation of V*(s) can be used to calculate the value of states, representing the expected cumulative returns associated with being in a given situation and subsequently following the best decision strategy throughout the air combat. The Bellman optimal equation constructed with the state action value function can also be used.

### 2.2. Air Combat Environmental Model

The defined air combat adversarial environment for the MDP is implemented as two simulators ${Simu}_{i},i\in \{Horizontal,Vertical\}$, where $\left(S_{next},R_{i}\right)={Simu}_{i}(S_{i},A_{i})$ and $A_{i}$ is the action of Agent i in state $S_{i}$ [44]. The simulator ${Simu}_{\left(i\right)}$ receives the action $A_{i}$ and then produces the next state $S_{i}$ and the reward $R_{i}$, where the state space $S_{i}$ consists of the coordinates (*x*,*y*,*z*), velocity v and acceleration Δ of the red and blue sides under the geographic coordinate system:(6)$$S=(x_{r},y_{r},z_{r},v_{x_{r}},v_{y_{r}},v_{z_{r}},\Delta_{x_{r}},{\Delta_{y_{r}},\Delta_{z_{r}},x}_{b},y_{b},z_{b},{v_{x_{b}},v_{x_{r}},v_{y_{b}},v}_{z_{b}},\Delta_{x_{b}},\Delta_{y_{b}},\Delta_{z_{b}},)$$

In the next state, the geometric position, the spatial positions of the tracker, and the target are updated after the input actions [45,46]. The action space in Horizontal space are discrete, and they are defined as three different actions: forward, left, and right. Again, action space Vertical is defined as three different actions: up, hold, and down. In addition, we specifically set rules on height for this simulation to match realistic scenarios so that, during training, if the tracker moves beyond the restricted height range, the simulator limits its further descent or ascent and then receives a new movement [47]. We define rewards $R_{i}$ for the corresponding environment,i∈{Horizontal,Vertical}. The role of the reward function is to encourage the tracker to continuously track the movement of the target. It is defined as follows:(7)$$R_{i}=\omega_{1}SOT(f_{target},f_{state})$$
where $\omega_{1}$ is a parameter and is a positive parameter, $f_{target}$ represents the real position and velocity of the target, and fstate represents the current position and velocity of the tracker. SOT represents the status of tracking between $f_{state}$ and $f_{target}$. The DQN [14] algorithm is applied to the learning of each agent in the simulation. It learns an optimal control policy $\pi_{i}:S_{i},G_{i}\to A_{i},i\in \{Horizontal,Vertical\}$.

The horizontal position between the aircraft and the target is indicated by $C=(\varphi_{u},D)$, where D is the azimuth of the aircraft and the distance between the two aircraft, respectively. Figure 2 depicts the position of the tracker relative to the target. The subscripts 𝑢 and 𝑡 indicate the tracker aircraft and the target, respectively, and $\varphi_{t}$ indicates the azimuth of the tracker relative to the target.

The most important platform capability in air combat countermeasures training systems is flight capability, so this paper presents designs for a set of motion models to model the aircraft platform, which mainly reflect the flight trajectory under the limitation of aircraft flight performance. The six degrees of freedom for aircraft require consideration of the warplane as a rigid body, the complexity of the aircraft structure, and its longitudinal coupling. Here, a three-degrees-of-freedom model is used, ignoring the aircraft as a rigid body, treating it as a mass, and assuming that the flight control system can respond accurately and quickly to form a maneuver trajectory. The core of the maneuvering decision problem is the rapid generation of the dominant maneuvering trajectory, and the aircraft three-degrees-of-freedom model can meet the solution requirements. The aircraft three-degrees-of-freedom model includes a mass point model of the aircraft platform and a dynamics model; the control model is shown in Figure 3. The specific models are
(8)$$\left\{\begin{array}{l} \dot{V}=g(n_{x}-\sin(\theta ))\\ \dot{\theta }=\frac{g(n_{z}cos(\gamma )-cos(\theta ))}{V}\\ \dot{\varphi }=\frac{g(n_{z}sin(\gamma ))}{Vcos(\theta )}\\ \dot{X}=Vcos(\theta )cos(\varphi )\\ \dot{Y}=Vcos(\theta )sin(\varphi )\\ \dot{Z}=-Vsin(\theta ) \end{array}\right.$$
where 𝑥, 𝑦, and 𝑧 denote the position of the aircraft in the geographic coordinate system; V is the flight speed; θ is the velocity inclination angle, i.e., the angle between the velocity direction and the horizontal plane, with upward as positive; φ is the heading angle, i.e., the angle between the velocity direction on the horizontal plane and the due north direction, with clockwise as positive; and where it is assumed that the velocity direction is always in line with the direction in which the nose is pointing, i.e., the angle of attack and the sideslip angle are zero.

θ, φ, γ denote the trajectory inclination angle, track deflection angle, and roll angle, respectively; $n_{x}$ and $n_{z}$ denote the tangential overload along the velocity direction and the normal overload in the vertical velocity direction, respectively; and 𝑔 is the gravitational acceleration. In the above equation, the first three terms are the mass kinematics model and the last three terms are the aircraft dynamics model; the state variables include 𝑥, 𝑦, 𝑧, θ, φ, γ, 𝜓, and V; the control variables include $n_{x}$, $n_{z}$, and γ. Because an ideal mass point model is used in this study, the flight performance, control inputs, and control response parameters are restricted to make the trajectory and maneuvers of the target aircraft reasonable. Specifically, the values of $n_{x}$ and $n_{z}$ are limited to within 2 g and 5 g, respectively.

## 3. Hierarchical Reinforcement Learning Design

In this paper, we propose a hierarchical reinforcement learning training framework that comprises two parts: environment design and framework building. The purpose of the environment design is primarily to define the input and output state data available to the agent and the reward functions that can be obtained, and the framework building is primarily to establish the corresponding hierarchical network structure, realize the reward mapping corresponding to the course learning, and design the optimization algorithm and the corresponding training strategy.

### 3.1. Geometric Hierarchy in the Aircombat Framework

We formulate the intelligent body motion decision for a 3D air combat 1V1 confrontation as a Markov decision process (MDP), supplemented by a goal state G that we want the two agents to learn. We define this MDP as a tuple (*S*, *G*, *A*, *T*, μ), where *S* is the set of states, *G* is the goal, *A* is the set of actions, and *T* is the transition probability function. In this paper, a hierarchical reinforcement learning-based approach called a hierarchical reinforcement learning framework in geographic coordination for air combat, referred to as HRL-GCA, is used to build a shared multilevel structure. The method uses a technique called meta-learning, which learns from a set of tasks and applies this knowledge to new tasks. The algorithm can effectively build a shared multilevel structure, thus improving learning efficiency.

As shown in Figure 4, the global state *S* is a geometric representation of the tracker and target aircraft in a 3D simulated air combat scenario, including the positions S = (x, y, z) and velocities *v* = (vx, vy, vz) of both aircraft. At the beginning of each episode of each state $s_{i}$ in the MDP, for a given initial state $s_{0}^{}$ and target $g_{i}$, the solution to the sub-policy ω is a control policy $\pi_{i}$: $S_{i}$, $G_{i}$ → $A_{i}$ that maximizes the following value function:(9)$$v_{\pi_{i}}\left(s_{i},g_{i}\right)≔E_{\pi_{i}}[\sum_{t=0}^{\infty }\mu i_{t}R_{t}^{i}|s_{0}^{i}=s_{i},g_{i}=G_{i}]$$

An agent consists of an algorithm that updates a parameter vector (θ,ω) defining a stochastic policy $\pi_{\theta ,\omega }(s|a)$, where theω parameter is shared among all sub-policies, whereas the *θ* parameter is learned for each senior policy starting from zero, encoding the state of the learning process on that task. In the considered setup, an MDP is first sampled from the $P_{S}$, the agent is represented by the shared parameter ω and the randomly initialized *θ* parameter, and the agent iteratively updates its *θ* parameter during the T-step interaction with the sampled MDP. The objective of the HRL-GCA is to optimize the value contributed by the sub-policy over the sampled tasks:(10)$$V={\mathrm{maximize}}_{\theta }E_{S\sim P_{S},k=0,\ldots ,T-1}[v_{\pi }\left(s_{k},g_{k}\right)]$$
where π consists of a set of sub-policies $\pi_{1}$, $\pi_{2}$, …, $\pi_{N}$, and each sub-policy $\pi_{i}$ is defined by a subvector $\omega_{n}$. The network constructed by the parameter θ works as a selector. That is, the senior policy parameterized by θ selects the most appropriate behavior from index nϵ{1,2,…,N} to maximize the global value function V.

### 3.2. Reward Shaping

The senior action reward is used to train senior behaviors, which guide the sub-action to make further behavioral decisions. We take inspiration from the Meta-Learning Shared Hierarchies architecture to train the sub-policy independently, solidify its parameters, and then train senior action adaptively. Our approach is similar to Alpha-Dogfight [48], but we differ in that we implement further layering in the behavioral layer and map global rewards to local rewards by transformations under geographic coordination, and experimental results demonstrate that performance in the behavioral layer is further enhanced.

### 3.3. Senior Policy Reward

The senior policy performs discrete actions at a frequency five times lower than the sub-policy, which is 1 Hz and is trained using the same DQN as the sub-policy. The state space of the senior policy differs from that of the sub-policy, which is described in detail later in this paper. The reward for senior policy is given by
(11)$$r_{total}={\alpha r}_{angle}+{\beta r}_{dis}$$
where α and β are positive parameters and α+β=1.

Firstly, the angle reward $r_{angle}$ can help the model learn how to control the angle of the aircraft toward the target, and $\varphi_{u}$ is related to the limits of the detection angle of the airborne radar and the off-axis angle of the missile. Specifically, the attack advantage increases the closer $\varphi_{u}$ is to the desired angle, and $r_{angle}$ reaches its maximum when $\varphi_{u}$ = 0°, i.e., when the velocity is aligned with the target:(12)$$r_{angle}=e^{\left(-abs({f(\varphi }_{u})-{f(\varphi }_{t}))/180\right)}$$

Secondly, the distance redirection $r_{dis}$ is designed based on the distance between the aircraft and the target, which can help the model learn how to control the position of the aircraft to achieve a reasonable position about the target. Specifically, the smaller the distance *D* between the aircraft and the target, the higher the $r_{dis}$ value:(13)$$r_{dis}=e^{\left(-abs(f(D))/100\right)}$$

We used the above rewards for the initial training, and then in subsequent experiments, for comparison with other models, we adjusted the design of the reward to achieve the same state as the baseline. A description of how the three model rewards are adjusted in this paper can be found in Appendix A.

### 3.4. Sub-Policy Reward

However, the objective of this paper requires the mapping of rewards to the two subtask spaces, and we redistribute rewards for Agent 1 and Agent 2 via transformation in the geometric space. Because Agent 1 and Agent 2 are mainly implemented in two planes of control, as shown in Figure 3, $r_{total}^{}$ is achieved by mapping $\varphi_{u}$ and *D* to the x-y and x-z planes using the function δ to reconstruct the $G_{i}$.
(14)$$\begin{array}{c} r_{total}^{h}\left({\delta \left(\varphi_{u}^{}\right)}^{h}{,\delta \left(\varphi_{t}^{}\right)}^{h},{\delta \left(\overset{⃑}{D}\right)}^{h}\right)=\alpha_{h}r_{angle}^{h}\left({\delta \left(\varphi_{u}^{}\right)}^{h}{,\delta \left(\varphi_{t}^{}\right)}^{h}\right)+\beta_{h}r_{dis}^{h}\left({\delta \left(\varphi_{u}^{}\right)}^{h},{\delta \left(\overset{⃑}{D}\right)}^{h}\right)\\ r_{total}^{v}\left({\delta \left(\varphi_{u}^{}\right)}^{v}{,\delta \left(\varphi_{t}^{}\right)}^{v},{\delta \left(\overset{⃑}{D}\right)}^{v}\right)=\alpha_{v}r_{angle}^{v}\left({\delta \left(\varphi_{u}^{}\right)}^{v}{,\delta \left(\varphi_{t}^{}\right)}^{v}\right)+\beta_{v}r_{dis}^{v}\left({\delta \left(\varphi_{u}^{}\right)}^{v},{\delta \left(\overset{⃑}{D}\right)}^{v}\right) \end{array}$$
Here,
(15)$$r_{angle}^{h}\left({\delta \left(\varphi_{u}^{}\right)}^{h}{,\delta \left(\varphi_{t}^{}\right)}^{h}\right)=e^{\left(-abs(f(\varphi_{u}^{h})-f(\varphi_{t}^{h}))/180\right)},$$
(16)$$r_{dis}^{h}\left({\delta \left(\varphi_{u}^{}\right)}^{h},{\delta \left(\overset{⃑}{D}\right)}^{h}\right)=e^{\left(-abs(D^{h}*{f(\varphi }_{u}^{h})/100\right)},$$
(17)$$r_{angle}^{v}\left({\delta \left(\varphi_{u}^{}\right)}^{v}{,\delta \left(\varphi_{t}^{}\right)}^{v}\right)=e^{\left(-abs(f(\varphi_{u}^{v})-f(\varphi_{t}^{v}))/180\right)},$$
(18)$$r_{dis}^{v}\left({\delta \left(\varphi_{u}^{}\right)}^{v},{\delta \left(\overset{⃑}{D}\right)}^{v}\right)=e^{\left(-abs(D^{v}*f(\varphi_{u}^{v}))/100\right)}.$$

The redistribution of rewards is achieved by the function δ. The δ function is a spatial projection operator that maps reward elements $\varphi_{u}^{}$, $\varphi_{t}^{}$, and $\overset{⃑}{D}$ to the x-y and x-z planes, respectively. This ensures that the reward functions $r_{total}^{h}$ and $r_{total}^{v}$, which are used for training in the x-y and x-z planes, have the same expression. However, their auto-covariates are the result of the projection through the δ posterior: $\varphi_{u}^{h}$, $\varphi_{t}^{h}$, $D^{h}$, and $\varphi_{u}^{v}$, $\varphi_{t}^{v}$, $D^{v}$, respectively, as detailed in Appendix B. Of these, reward $r_{total}^{h}$ allows the tracker to follow the target better on the x-y surface, and reward $r_{total}^{v}$ is used to suppress the altitude difference and, as much as possible, encourage the aircraft to be at the same altitude level as the target at high altitude. In addition, in this paper, the treatment in Equation (7) is also applied to its rewards in the comparisons with other baselines.

### 3.5. Hierarchical Training Algorithm

In this paper, a course learning approach is used for hierarchical training; the definition of the algorithm is detailed in Appendix C, and the policy network is trained to interact with the environment at a frequency of 10 Hz. The same observation space is used for both policies.

We then explore cooperative learning between Agent 1 and Agent 2 in the training of horizontal control and height control policies. In each iteration of the learning, firstly, Agent 1 moves the tracker on the x-y surface of the 3D geographic coordination scenes; secondly, the next state $s_{t+1}^{1}$ and the intermediate state $s_{t}^{2}$ update after action $a_{t}^{1}$; and thirdly, Agent 2 moves the tracker on the x-z surface. The next state $s_{t+1}^{2}$ updates after $a_{t}^{2}$.

Initial conditions: These initial conditions are divided into tracking targets that start moving from different positions and take different forms of motion in the height and horizontal planes. Concerning stochastic multistep payoffs, for time–distance learning, multistep payoffs tend to lead to faster learning when appropriately tuned for the number of steps to be used in the future. Instead of tuning a fixed value, we define the maximum number of steps in the future and uniformly sample the maximum value. A common expression for future value is
(19)$$Q\left(S^{\prime },A\right)\leftarrow Q\left(S,A\right)+\partial \left(R+\tau \underset{a^{\prime }}{\max}Q(S^{\prime },a^{\prime })-Q\left(S,A\right)\right)$$

The tactical objective of the horizontal plane tracking subtask is to enable the tracker to continuously track the target aircraft in the x-y plane. Formally, motion in the x-y plane is achieved by outputting horizontal motion, successive horizontal left turns, and successive horizontal right turns at each simulation step with a constant steering speed of 18°/s. The initial and termination conditions for the x-y subtasks are designed as shown in Figure 2. The tactical objective of the altitude tracking subtask is to enable the tracker to follow the target aircraft consistently at altitude. The mission can start in any state. Formally, motion in the x-z plane is achieved by outputting horizontal motion, continuous climb, and continuous descent in each simulation step, with a constant climb and descent rate of 20 m/s.

This in turn contains one output, namely, the value of ${Q(s,x}_{i})$. The activation function is the logsoftmax function: (20)$${Q(s,x}_{i})=\left(x_{i}-x_{m}\right)-log(\sum_{j=0}^{n}e^{x_{j}-x_{m}})$$
and Equation (20) directly outputs the value of each action using the logsoftmax nonlinear function, where $x_{m}$ is the largest element of $X=(x_{1},x_{2},\ldots x_{n})$.

### 3.6. Hierarchical Runtime Algorithm

In the hierarchical runtime algorithm, we explore the cooperation of Agent 1 and Agent 2 in a 3D simulated air combat situation. The algorithm is defined in detail in Appendix D. In each iteration of learning, firstly, Agent 1 moves in the x-y plane of the 3D air combat scenario; secondly, the next state $s_{t_{1}+1}^{}$ and intermediate state $s_{t_{2}}^{}$ are updated after action; and thirdly, Agent 2 moves up or down in the x-z plane. The next state $s_{t_{2}+1}^{}$ is updated after $a_{t_{2}}^{}$.

For each action 𝑚𝑖, a minimum period 𝑡 = 1500 milliseconds and a maximum period 𝑢𝑖 = 4 milliseconds are set. When the reinforcement learning intelligence outputs the action 𝑚_𝑖_ (including the stop action) at moment T, it starts to execute 𝑚_𝑖_ if no action is executed at the previous moment T − 1. If moment T – 1 performs action 𝑚_𝑗_ and the execution time is greater than or equal to 𝑡, then at moment T, the agent will be allowed to execute 𝑚_𝑖_ to replace the action 𝑚_𝑗_, otherwise not. If moment T – 1 performs action 𝑚_𝑗_ and the execution time is less than 𝑡_𝑗_, then the output behavior 𝑚_𝑖_ at moment T is ignored. When the reinforcement learning intelligence outputs no behavior (which is not the same as the stopping behavior) at moment T, if the previous moment T – 1 performed the behavior 𝑚_𝑘_ and the execution time is greater than or equal to 𝑢𝑖, then the execution of the no-behavior starts; otherwise, the execution of the behavior 𝑚_𝑘_ continues. The setting of the min-max period can to some extent prevent incorrect behavior of the flight unit.

## 4. Results

### 4.1. Experimental Environment Setup

The experiments in this paper use a hierarchical reinforcement learning framework to solve the problem in an air combat simulation environment. The hardware environment used in the experiments is an Intel Core i7-8700K CPU, 16 GB RAM, and an NVIDIA GeForce GTX 4090 Ti graphics card. The size of the 3D space in the experiment is 100 km × 100 km × 10 km; there are 20,000 × 480 s training episodes for each model; and the actual data sampling frequency is 10 HZ. The experimental results show that the performance of the algorithm improves significantly after the training of 20,000 episodes.

### 4.2. Performance Metrics during Training and Validation

To select the best-performing agent, we create an evaluation metric to compare the training results of various methods. The qualitative and quantitative results demonstrate the usefulness of our proposed model. The tracking performance of the tracker is evaluated when the target is moving at 0–180° relative north in an air combat environment. For comparison, we trained 2400 episodes for each angle type, for a total of 11.5 × 10^6^ simulation steps, and tested 500 samples for the corresponding angle types.

The meanings of each indicator are as follows: miss distance represents the average distance between the tracker and the target during the entire tracking process; miss angle represents the average track angle $\varphi_{u}$ between the tracker and the target during the entire tracking process; approach time represents the time taken to approach the target for the first time to a certain distance; hold distance time is the length of time that the tracker stays within a certain distance of the target; hold angle time is the time that the tracker stays within a certain angle of the target; and cost time refers to the time spent by each strategy model when outputting the current action command.
(21)$$P\left(Miss\;Distance\right)=\frac{acc(total\;dis)}{t_{epoch\;time}}$$
(22)$$P\left(Miss\;Angle\right)=\frac{acc(track\;angle\;\varphi_{u})}{t_{epoch\;time}}$$
(23)$$P\left(Approach\;time\right)=\tau_{TOA}-\tau_{0}$$
(24)$$P\left(Hold\;distance\;time\right)=\frac{\tau (dis\leq \sigma )}{\tau (epoch\;time)}$$
(25)$$P\left(Hold\;angle\;time\right)=\frac{\tau (angle\leq \partial )}{\tau (epoch\;time)}$$
(26)$$P\left(Cost\;Time\right)=\tau (\theta )$$

### 4.3. Validation and Evolution of the Hierarchical Agents

In this experiment, we reproduce the models and algorithms in three papers [9,15,49], and apply the hierarchical reinforcement learning framework established in this paper to learn and train them, respectively, while mapping the reward functions shaped in the three papers in the corresponding sub-state spaces; then, in the air combat environment established in this paper, different models are compared in the same test scenarios, and the performance of the three original models is compared with that of the models after applying HRL. We use the benchmark performance comparison method proposed in Section 4.2 to compare the models proposed in the paper, as shown in Table 1. Models 1, 2, and 3 denote the performance of the three models. The experimental results show that the HRL-GCA proposed in this paper can achieve higher scores in all three dimensions under the six test metrics compared with the other three models: the miss distance, miss angle, and approach time decreased by an average of 5492 m, 6.93 degrees, and 34.637 s, respectively, and the average improvement of angle maintenance and distance maintenance time is 8.13% and 16.52%, respectively. Of the other models, Model 2 has the highest hold distance and hold angle time with percentages of 41.12 and 15.44, respectively. In addition, the HRL-GCA model can converge faster and achieve higher accuracy in the training process. Therefore, we conclude that HRL-GCA demonstrates better performance in this experiment.

As shown in Table 1, the implementation of HRL models results in a 40–50% increase in runtime compared to the baseline models. This can be attributed to the fact that HRL models involve more complex computations and require more processing time. This is mostly because HRL incorporates several learning layers. Consequently, the HRL will execute over two extra neural networks in addition to the base models. 

Notwithstanding, we consider the time cost to be acceptable based on the comparative results presented in Table 1. For instance, Model 2 benefited from HRL improvement, requiring only a minimum of 87.56 s for Approach Time and making approximately 65 decisions for the approach to the target. In contrast, the corresponding model without HRL improvements required 137.06 s for Approach Time, making about 145 decisions for the approach to the goal. The HRL-improved model achieves its goal with only 65 decisions compared to the original model’s 145, resulting in a 55% improvement in decision-making efficiency. This increase in efficiency of the HRL-improved model (55%) offsets the additional time overhead required to execute the model (43.40%).

Furthermore, as an example, Model 2 shows improved Hold Distance Time and Holding Angle Time by 16.33% and 8.24%, respectively, after implementing HRL. Furthermore, compared to the model without HRL improvement, the distance and angle tracking stability are enhanced by 65% and 114%, respectively. In summary, although the computation time spent increased by 43.40%, the HRL improvement resulted in a 55% increase in decision efficiency within the same timeframe. The distance and angle tracking stability also increased by 65% and 114%, respectively. Therefore, this improvement is deemed reasonable.

## 5. Discussion

### 5.1. Trajectory of Air Combat Process

As shown in Figure 5, Figure 6, Figure 7 and Figure 8, we deploy the algorithm of this paper in a typical air combat scenario and compare its tracking of the target aircraft with a model that does not use this algorithm. During air combat, continuous tracking of the target aircraft in a given scenario is necessary to shoot it down. In the test cases, the target aircraft maneuvers continuously at altitude and moves away from the tracker by turning away from it, as seen in the 3D and 2D tracking trajectories, but the tracker ensures continuous alignment with the target in both altitude and direction. In contrast, the use of the other model fails to achieve continuous tracking of the target in either direction or altitude. Furthermore, the red dashed line in Figure 6 and Figure 8 shows the desired tracking trajectory for the target.

In our experiments, we use a hierarchical reinforcement learning framework to optimize and enhance the vehicle tracking trajectories. The trajectories in Figure 9 show the tracking states of the modified model 2 based on HRL and the model set out in this paper in the XY plane, XZ plane, and XYZ 3D space, respectively. Of these, in Figure 10, the red line is the tracking flight, the blue line is the tracked flight, and the number represents the flight trajectory sequence of both flights. The experimental results show that the use of the hierarchical reinforcement learning framework can effectively improve the accuracy and stability of aircraft tracking trajectories and can effectively reduce their deviation. It is found that Model 3 is more sensitive to the weighting parameters α,β in Equation (11) and has the best test results when the two reward ratios in the original paper are set to 0.5, 0.5. Irrespective of the rewards in Models 1, 2, and 3 or the rewards used in this paper, in Figure 11 and Figure 12, the tracking performance of a single network simultaneously controlling the motion of the horizontal plane and the motion of the height layer is inferior to that of multiple networks controlling them separately. In addition, the experimental results show that the use of the reinforcement learning method can effectively improve the accuracy of aircraft tracking trajectories, thus improving the timeliness of target tracking.

### 5.2. Training Process

The analysis of the experimental results in this paper shows that we can compare the changes in reward and loss of Models 1, 2, and 3 with the HRL-GCA model during the training process. From the experimental results, the reward and loss of HRL-GCA converge as the episodes increase and reach their optimal state after stabilization. In Figure 13, from the change in reward, our research model reward reaches its maximum at episode 21, whereas the rewards of the three standard models still show large fluctuations at episode 21, indicating that the reward of our research model has better convergence performance. Figure 14 illustrates the loss parameters during training after normalization. From the change in loss, the loss of our research model reaches its minimum at episode 592, whereas the losses of the three standard models still show a large fluctuation at episode 592, indicating that our research model has a better convergence performance of loss. 

In summary, Figure 13 and Figure 14 show that our research model has the best convergence performance during the training process, as well as an optimal state after stabilization. Therefore, when introducing a new sub-policy, the framework in this paper can achieve fast adaptation in training and learning for the corresponding task. Furthermore, in Figure 13, Model 3 and the proposed model decrease significantly after the first peak around episode 21; such behavior makes these models inferior to Model 2. This is mainly due to overfitting. From Equations (9) and (10) in Section 1, it can be deduced that each sub-policy $\pi_{i}$ can quickly learn its corresponding subvector $\omega_{n}$ after the initial learning phase, but $\omega_{n}$ learns only a tiny portion of the state space, and it needs to further learn the θ corresponding to the selector to maximize the global value function V. 

In addition, the θ corresponding to the selector can be learned from the $\omega_{n}$ subvector, but due to the large amount of training data required to train θ, as $\pi_{\theta ,\omega }(s|a)$ performs stochastic exploration, the variance is large, resulting in a decrease in $v_{\pi }\left(s_{k},g_{k}\right)$ until $S\sim P_{S}$ accumulates sufficiently to train a valid θ parameter. Furthermore, model 2 can maintain a more stable exploration-utilization capability throughout the training process, but the model proposed in this paper has a higher final reward value compared to the other models due to a better memory of what has been learned. Finally, we use black dashed lines in Figure 13 and Figure 14 to show the variation of reward and loss with the training process in the ideal case.

## 6. Conclusions

The hierarchical reinforcement learning framework in geographic coordination for air combat proposed in this paper trains two types of neural networks using distance reward, angle reward, and a combination of both to control the vehicle in multiple dimensions. The model has achieved good results in tracking targets in multiple dimensions. In thousands of tests, it achieved an average improvement of 8.13% in angle tracking and 16.52% in distance tracking over the baseline model, demonstrating its effectiveness. However, the model has limitations, especially in complex environments or when the goal is to perform complex maneuvers, and it is not yet able to achieve optimal control. Future research should focus on improving the tracking performance in such scenarios, along with exploring additional reward functions to improve stability and accuracy. Furthermore, numerous challenges remain, such as addressing two-agent game combat and extending to 2v2 and multi-agent combat scenarios in air combat pair control, which warrant further exploration. In addition, the application potential of this model in other real-world scenarios should be investigated.

## Figures and Tables

**Figure 1 entropy-25-01409-f001:**
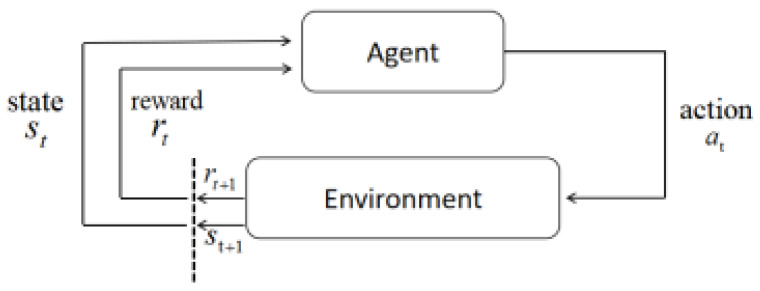
Markov process model.

**Figure 2 entropy-25-01409-f002:**
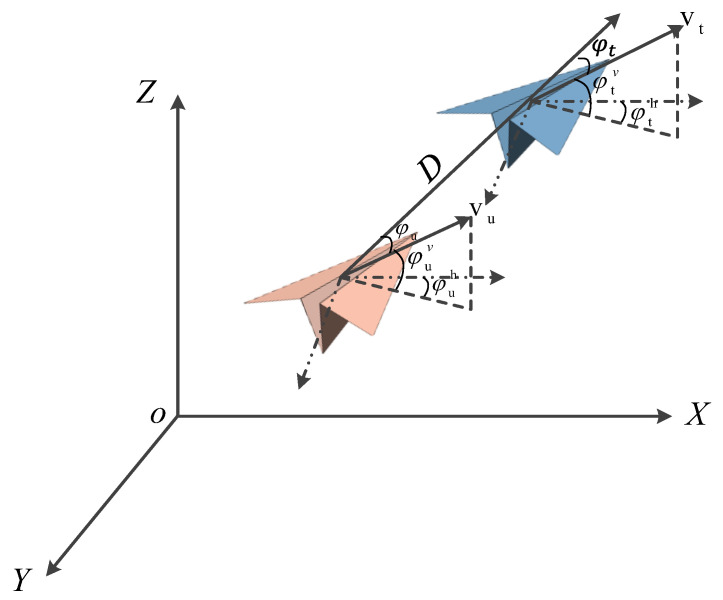
Vehicle situational model.

**Figure 3 entropy-25-01409-f003:**
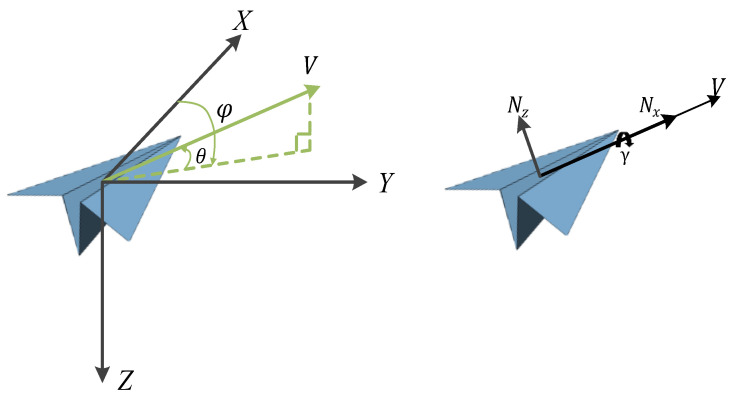
Vehicle control model.

**Figure 4 entropy-25-01409-f004:**
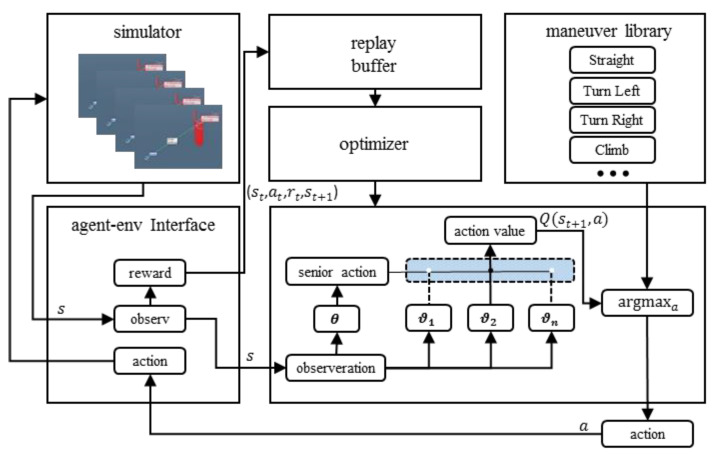
Model structure and training framework.

**Figure 5 entropy-25-01409-f005:**
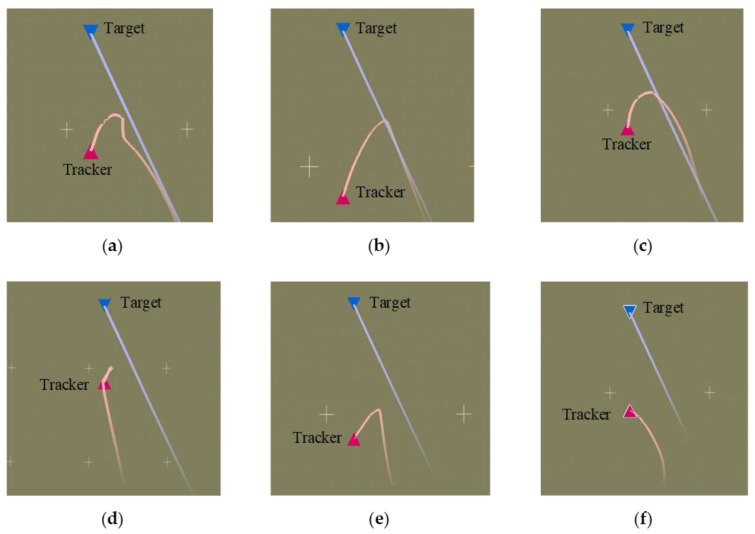
Angle tracking performance: comparison of models ((**a**–**f**) represents the horizontal tracking trajectory of model 1 with HRL framework, the horizontal tracking trajectory of model 2 with HRL framework, the horizontal tracking trajectory of model 3 with HRL framework, the horizontal tracking trajectory of model 1 without HRL framework, the horizontal tracking trajectory of model 2 without HRL framework, and the horizontal tracking trajectory of model 3 without HRL framework, respectively).

**Figure 6 entropy-25-01409-f006:**
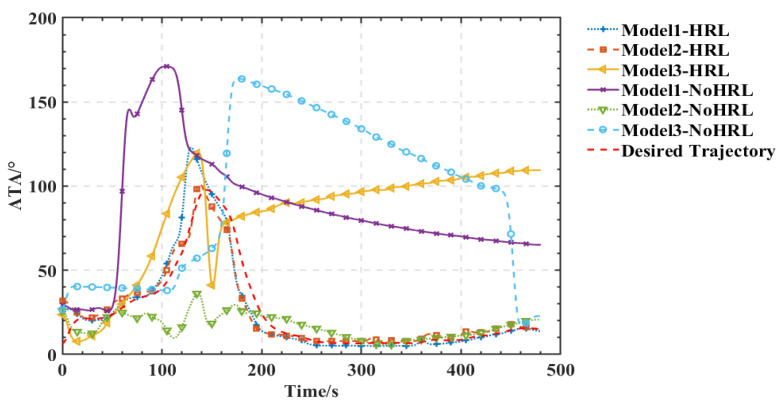
Comparison of angle tracking states of different models in the same scene.

**Figure 7 entropy-25-01409-f007:**
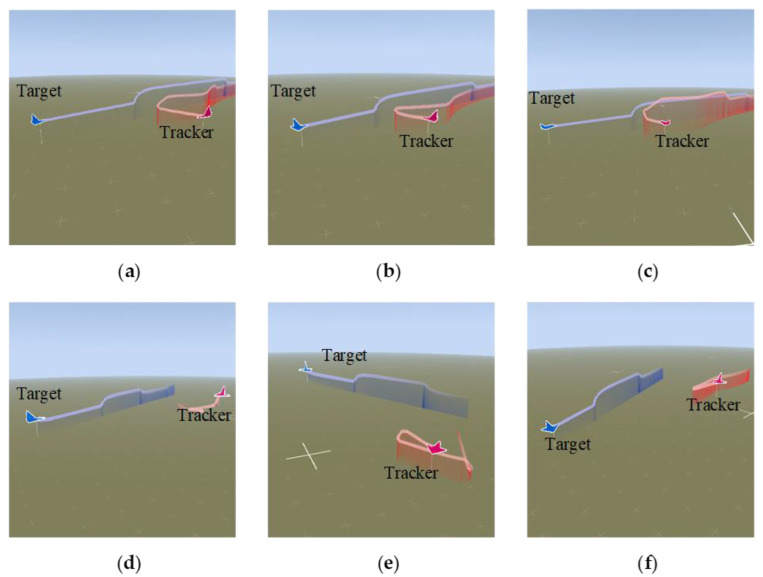
Comparison of the height tracking performance of the models ((**a**–**f**) represents the vertical tracking trajectory of model 1 with HRL frame, the vertical tracking trajectory of model 2 with HRL frame, the vertical tracking trajectory of model 3 with HRL frame, the vertical tracking trajectory of model 1 without HRL frame, the vertical tracking trajectory of model 2 without HRL frame, and the vertical tracking trajectory of model 3 without HRL frame, respectively).

**Figure 8 entropy-25-01409-f008:**
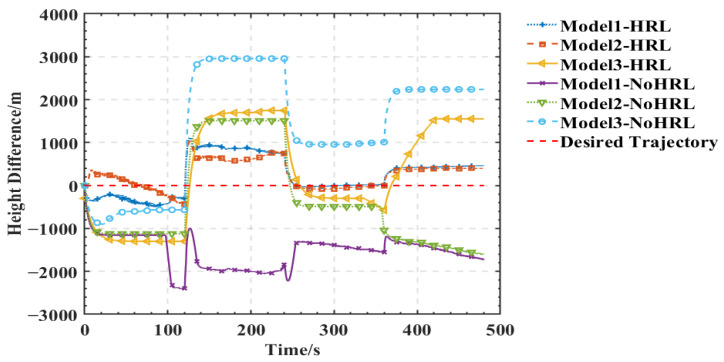
Comparison of different models in the same scene with height-tracked states.

**Figure 9 entropy-25-01409-f009:**
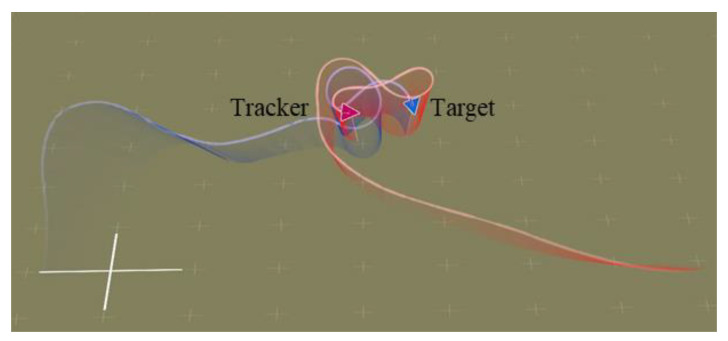
Tracking the trajectory of the HRL modified model 2 agent.

**Figure 10 entropy-25-01409-f010:**
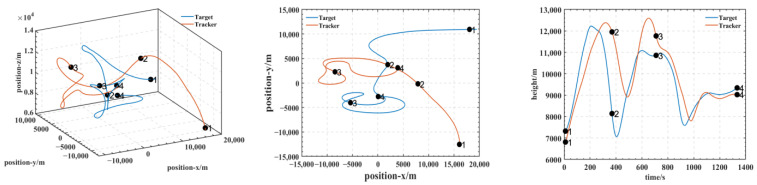
Three-dimensional and 2D trajectories when the target and tracker are using the HRL agent.

**Figure 11 entropy-25-01409-f011:**
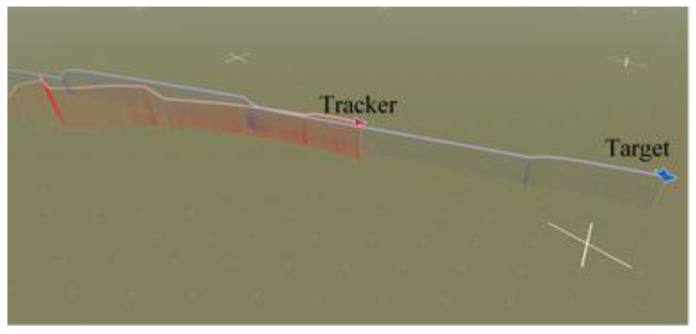
Tracking the trajectory of the HRL-free model 2 agent.

**Figure 12 entropy-25-01409-f012:**
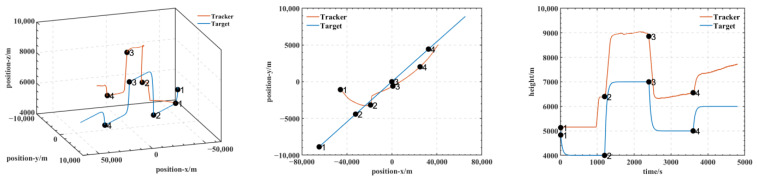
Three-dimensional and 2D trajectories when the target and tracker are not using the HRL agent.

**Figure 13 entropy-25-01409-f013:**
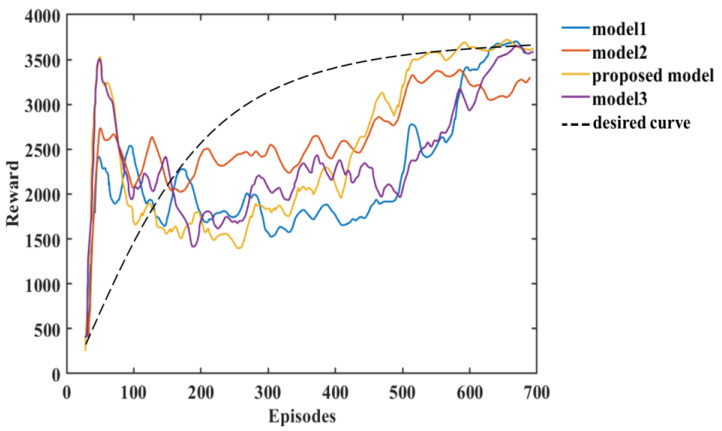
The relationship between the reward and episodes of the models.

**Figure 14 entropy-25-01409-f014:**
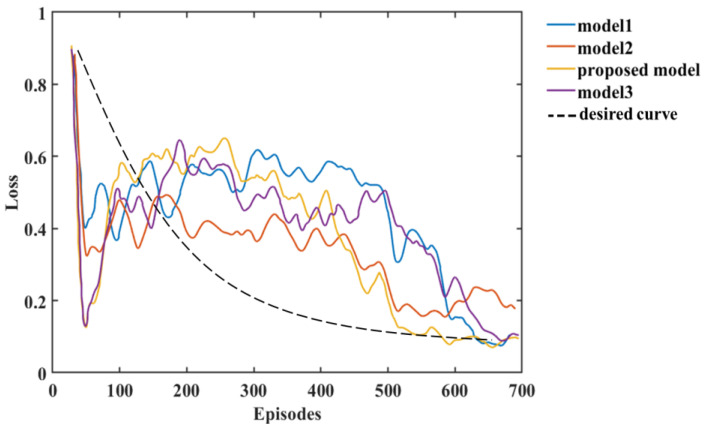
The relationship between the loss and episodes of the models.

**Table 1 entropy-25-01409-t001:** Comparison of experimental results with and without the HRL framework.

**Reward Type**	**Miss Distance (m)**			**Miss Angle (°)**			**Approach Time (s)**		
	**without hrl**	**with hrl**	**Ratio of Decrease**	**without hrl**	**with hrl**	**Ratio of Decrease**	**without hrl**	**with hrl**	**Ratio of Decrease**
Model 1	44,378.83 (±4020.2)	38,900.41 (±3778.3)	12.34%	32.56 (±5.73)	25.71 (±1.46)	21.04%	141.99 (±19.89)	101.459 (±20.09)	28.54%
Model 2	41,696.70 (±3692.7)	35,797.28 (±3494.9)	14.14%	31.68 (±2.66)	28.726 (±8.39)	9.32%	137.06 (±21.32)	87.56 (±17.83)	36.11%
Model 3	43,526.82 (±2332.4)	38,427.20 (±2371.3)	11.71%	36.69 (±6.93)	25.69 (±6.41)	29.98%	102.89 (±25.18)	89.01 (±18.66)	13.49%
**Reward Type**	**Hold Distance Time (%)**			**Hold Angle Time (%)**			**Cost Time (ms)**		
	**without hrl**	**with hrl**	**Ratio of Increase**	**without hrl**	**with hrl**	**Ratio of Increase**	**without hrl**	**with hrl**	**Ratio of Increase**
Model 1	17.15% (±6.39%)	33.14% (±3.87%)	15.99%	5.23% (±0.22)	11.38% (±0.99%)	6.15%	0.97	1.427	47.11%
Model 2	24.79% (±4.99%)	41.12% (±4.67%)	16.33%	7.2% (±0.63%)	15.44% (±1.67%)	8.24%	0.94	1.348	43.40%
Model 3	19.30% (±4.61%)	36.56% (4.51%)	17.26%	4.06% (±1.07%)	14.07% (±1.72%)	10.01%	0.91	1.410	54.95%

## Data Availability

The data used to support the findings of this study are available from the corresponding author upon request.

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
