# Peer review of "Hierarchical Reinforcement Learning Framework in Geographic Coordination for Air Combat Tactical Pursuit"

_entropy, 2023, doi:10.3390/e25101409_

Round 1
Reviewer 1 Report
General comments:
This paper proposes a method to improve the decision-making process in aerial combat training by decomposing the 3-dimensional flight dynamics into two 2-dimensional ones and integrating a hierarchical reinforcement learning approach. The process can be of great interest to researchers studying aerial combat and reinforcement learning.
Although it is mentioned in the conclusion that the tracking ability may be limited if the target performs complex maneuvers, it is unclear how the effectiveness the proposed approach would degrade as the complexity increases. In the test cases, the target flies either along a straight path (Figs. 5 and 7) or performs only 1-2 turns (Fig. 10 and 12), which are rather simple maneuvers in aerial combat. The authors should clarify the limitations of the proposed approach using quantitative measures, e.g. maximum g forces the target can perform, etc.
While the focus of this paper is reinforcement learning, the authors should include some discussions on whether the resulting strategy (action sequence) can be performed within the capability of the aircraft-pilot system. In other words, the proposed approach should verify if each action requires excessive thrust that the aircraft cannot generate or requires excessive g loads that the pilot cannot withstand. Otherwise, the unconstrained optimal strategy could potentially be infeasible.
In addition, please revisit the formatting of equations and variables to make them easier to understand.
Specific comments:
Pg. 3, 3 lines above Eq. 3: It appears that the right hand side of the inequality should be V^{\pi}(s).
Pg. 4, Eq. 6: Please clarify if v and \Delta are supposed to be scalars or 3-d vectors, since the position coordinates have been already decomposed into scalar x, y, and z.
Pg. 8, paragraph before Sec. 3.4: How is the design of the reward adjusted? What parameters or functional forms are adjusted?
Pg. 8, Eqs. 12, 13, 15-18: The function f is not defined in the context. If it is meant to be a generic form, the authors should clarify its characteristics, e.g., monotonicity, periodicity, etc.
Pg. 8, Eqs. 14-18: The function \delta is not defined in the context.
Pg. 10 paragraph before Sec. 3.6: What is “VALUE” here?
Same paragraph: Please disambiguate “this” as a subject in the paragraph.
Pg. 10 first paragraph in Sec. 3.6: Please double check if agent2 should move the airplane in the x-z plane as mentioned earlier in the manuscript.
Pg. 10 paragraph after Algorithm 2: How are the time span t and ui set?
Same paragraph: The same letter t appears to describe both the state (ordinal and non-dimensional) and the execution time (continuous and dimensional), which can be confusing. Consider replacing either one with another letter.
Same paragraph: “If moment t-1 performs action mj and the execution time is greater than or equal to tj, then the execution of mi is started.” – It appears that mi starts before mj is finished, so the two actions overlap. Please double check the statements in this and the next sentences.
Same paragraph: “The execution of the behavior continues k” – what is k here?
Pg. 11 first paragraph in Sec. 4.2: What are the “fields” here?
Pg. 11 second paragraph in Sec. 4.2: Please also define the miss angle.
Table 1: It is recommended to add a third column under each metric to highlight the improvements with HRL.
Table 1: Based on the increased cost time, does HRL require about 40-50% longer runtime compared to the existing models?
Figs. 5, 7, 10, and 12: Please label in the figures which marker is the tracker and which is the target, so readers do not have to search in the text for explanation.
Figs. 9 and 11: How do the cockpit views support the discussions? If they do not, please remove them.
Figs 10 and 12: There is only one pair of trajectories in each plot. Which model (1/2/3) does it correspond to?
Figs. 10 and 12: The four sequency numbers on each pair curves do not occur at the same time. For example, in Fig. 10, number 1 occurs at about 200 s in the blue trajectory but 300 s in the red trajectory. The authors should address such a misalignment.
Fig. 10: The altitude gap and the horizontal distance do not appear to approach zero after point 4. How does this figure support the discussions?
Fig. 10: In the first chart (x-y-z plot), the blue trajectory has y < 0 everywhere and the red trajectory has x > 0 everywhere, while in the second chart (x-y plot), the blue trajectory has y > 0 everywhere and the red trajectory has x < 0 everywhere. Please explain.
Fig. 12: In the first chart, the blue trajectory has x < 0 everywhere, while in the second chart it is in the region where x > 0. Please explain. It is difficult to tell if the red trajectory has a similar issue, but please double check.
Fig. 12: In the last chart (altitude vs time), why are the flight trajectory sequence numbers (1-2-3-4) decreasing in time?
Pg. 15 paragraph below Fig. 12: “… Model 3 was more sensitive to the weighting parameters” – what are the weighting parameters?
Same paragraph: “… when the four reward ratios in the original paper” – what are the reward ratios?
Pg. 16 first paragraph in Sec. 5.2: The first peak in the reward appears to occur at around episode 20 instead of 50.
Same paragraph: Why does the reward of Model 3 and the proposed model decrease significantly after the first peak around episode 20? Such a behavior makes these models inferior to Model 2 until around episode 400. Please provide some relevant discussions.
Same paragraph: There are discussions on the loss but without a supporting chart like Fig. 13. Please provide one in the manuscript to substantiate the argument.
There are some mixed use of present tense and past tense. It is recommented to consistently use present tense outside literature review.
Reviewer 2 Report
In this paper, the authors present the investigation of an air combat training structure based on hierarchical reinforcement learning. Results from computer simulations demonstrated that the proposed framework could achieve better performance than the baseline approach.
The paper presents interesting and relevant results, however the discussions of the results should be improved.
a- Authors should replace figures 6 and 8 with figures with lines with markers, only with different colors it is difficult to identify each line;
b- The line that represents the desired trajectory in figures 6, 8 and 13 must be included;
c- The algorithms can be transferred to an appendix section;
d- The results should be compared with the results of the cited references, I recommend including a comparison table;
c- Considering the requested corrections, a new conclusion must be provided.
Round 2
Reviewer 1 Report
Eqs. 14-18, regarding previous comment 5: The function \delta (lower-case) is still present without a definition. Please define it in the context.
Pg. 10, second line, regarding previous comment 12: Please check if the sentence should read “the execution of the behavior m_k continues.”
Regarding previous comment 16: The authors’ response in the cover letter could add a lot of value to the paper. Consider adding a subsection or a paragraph based on the response to address the trade-off between runtime and accuracy, etc. since other readers may raise the same question as I did previously.
Fig. 6: It is recommended to also use a marker for Model2-NoHRL to keep the line styles consistent with Fig. 8.
Fig. 7: There is a misplaced “Target” label in subfigure (a).
There are some very minor issues that can be easily resolved by further proofreading, e.g. Pg. 15 "In summary, Figure 13 and Figure 14 shows that..."
Reviewer 2 Report
The authors made all the requested corrections and clarifications, submitting a new version that can be accepted for publication in its current form.
Author Response
Special thanks to you for your good comments.